# Pemphigus during the COVID-19 Epidemic: Infection Risk, Vaccine Responses and Management Strategies

**DOI:** 10.3390/jcm11143968

**Published:** 2022-07-08

**Authors:** Xueyi Huang, Xiaoqian Liang, Jiao Zhang, Hang Su, Yongfeng Chen

**Affiliations:** Dermatology Hospital, Southern Medical University, Guangzhou 510091, China; hxy202207@126.com (X.H.); liangxq1102@163.com (X.L.); zgjojoyce@163.com (J.Z.); brilliantsh@163.com (H.S.)

**Keywords:** autoimmune bullous diseases, pemphigus, COVID-19, vaccination, Rituximab, management strategies

## Abstract

Pemphigus is a rare autoimmune blistering disease, involving potentially life-threatening conditions often requiring immunosuppression. Currently, the COVID-19 pandemic caused by severe acute respiratory disease coronavirus 2 (SARS-CoV-2) infection has become a global public emergency. Vaccines are the most effective defense against COVID-19 infection. However, in clinic, there are cases of new onset or flare of pemphigus following COVID-19 vaccination, where vaccines have manifested significantly desirable risk-benefit profiles for patients. Although Rituximab, as first-line therapy, may impair humoral immunity, pemphigus may not predispose to develop COVID-19 infection compared to a healthy population. Conversely, delay or interruption of immunosuppressants probably results in unfavorable clinical outcomes for disease progression. Overall, clinicians should encourage their patients to undergo the vaccination after a comprehensive assessment. The definite association between COVID-19 vaccination and pemphigus remains to be further elucidated. Herein, we provide an overview of the published studies to date on COVID-19 and pemphigus as well as the exploration of their complicated interplay. In addition, we discuss the management strategies for pemphigus patients in this special period, in an effort to more effectively establish a standard treatment paradigm for this particular patient group.

## 1. Introduction

Pemphigus, a heterogeneous group of IgG-mediated autoimmune bullous disorders, is characterized by acantholysis (the loss of epithelial cell adhesion) that induces fragile blisters and painful erosions on the skin and mucosa [1,2]. IgG autoantibodies, which are directed against the desmosomal adhesion proteins, mainly desmoglein 3 and desmoglein 1 in epidermal keratinocytes, cause the blisters and epidermal splitting due to the compromised cell adhesion. The incidence rate ranges from 0.5 to 50 per million population, hinging upon geographic region and ethnicity; and its onset typically lies in individuals between the ages of 40 and 60 [3]. As a chronic and relapsed/refractory autoimmune disease, patients may develop diverse complications involving dehydration, malnutrition, and secondary infection [4], even life-threatening septicaemia that is mostly secondary to epidermal *Staphylococcus aureus* infection [5].

Systemic glucocorticoids, plus immunosuppressive adjuncts such as azathioprine are conventional treatment regimens for pemphigus [6]. Preliminary research reflected that non-targeted immunosuppressants would predispose pemphigus patients to infection in a dose-dependent manner [4]. Rituximab, a chimeric murine/human monoclonal antibody targeting CD20+ B lymphocytes including desmoglein-specific CD20+ B-cells, has yielded demonstrably robust efficacy in severe and refractory cases of pemphigus. As well, it met its approval as a first-line therapeutics for patients with moderate to severe pemphigus respectively by the British Association of Dermatologists (in 2017) [7] and the European Academy of Dermatology and Venereology (in 2020) [8]. However, limitations are worth noting that the inhibition of B cells is irreversible and immune recovery would take months.

The sudden emergence of the COVID-19 pandemic is caused by severe acute respiratory syndrome coronavirus-2 (SARS-CoV-2), which has infected millions of individuals and drew great attention across the globe. SARS-CoV-2 is a single-stranded RNA enveloped virus comprising four crucial structural proteins, including spike (S) glycoprotein, membrane (M), envelope (E), and nucleocapsid (N) protein [9]. Particularly spike protein, which is capable of binding to the host cell surface membrane via receptor-binding domain (RBD) and mediates endocytosis, plays a pivotal role in startup in infection. Statistics have suggested that vaccines remain the most effective approach to controlling viral advance [10]. Nonetheless, the safety of the COVID-19 vaccine for pemphigus patients and the underlying vaccine-induced responses still meet uncertainty thus far; and the management strategies for pemphigus patients during the COVID-19 pandemic remain incompletely established. Herein, we present the evidence to date about the relationship between pemphigus and COVID-19 infection and vaccination, and discuss whether pemphigus status is positively correlated with infection risk, as well as further investigate the precise role of COVID-19 vaccine in vivo, in an attempt to reveal insights for management strategies for pemphigus patients during this pandemic.

## 2. Pemphigus and COVID-19 Infection

According to a retrospective cohort study involving 245 pemphigus patients with an average following-up time per person of > 10 years [11], the overall mortality among pemphigus patients is 2.4-fold that of healthy controls, mainly due to infection. Namely, compared with the general population, the autoimmune status of pemphigus does have a higher risk of mortality from infections, probably owing that the integrity of the skin barrier disrupts, which easily leads to pathogen colonization and immune dysregulation [12]. However, the research did not provide a distinct answer to whether pemphigus state is more prone to viral infections than the healthy population. On the other hand, drawing on the findings of Wang et al. [13], most conventional immunomodulators are associated with a higher risk of predisposing to infection, while there is insufficient evidence to define whether patients on systemic immunomodulators are more likely to develop COVID-19 infection. In the trial of Saleh et al. [14], 72 patients with pemphigus vulgaris were enrolled and no adverse events about febrile syndrome, respiratory failure, or death due to COVID-19 were reported. A systematic review encompassing 732 autoimmune bullous diseases (AIBD) patients (211 pemphigus, 112 pemphigoid, 409 not specified) suggested that approximately 70 (9.5%) patients showed suspected symptoms of infection, among whom 16 (2.1%) were finally diagnosed. Six (0.8%) patients required hospitalization, and 3 (0.4%) died of COVID-19, all of whom were fragile, elderly, and comorbid patients [15]. The aforementioned results concurrently indicate that pemphigus seems to be not responsible for higher risk of COVID-19 infection. With regard to immunomodulatory treatment, the available data of previous coronaviruses reports reveal that immunosuppressed status did not negatively influence AIBD patients significantly [16]; hence long-term immunotherapy for patients does not constitute an independent risk factor. Prior research has characterized potential risk parameters of COVID19-associated mortality which essentially include advanced age, male sex, and disease comorbidities such as hypertension, diabetes, and obesity [17,18]. In another population-based cohort study which recruited 1845 pemphigoid patients and 1236 pemphigus patients, it was found that the risk of COVID-19-associated mortality in pemphigoid patients was higher than that in healthy individuals (hazard rate [HR], 2.82; 95% confidence interval [CI], 1.15–6.92; *p* = 0.023); in contrast, there is no significant difference between patients with pemphigus and their controls (HR, 1.33; 95%; CI, 0.15–11.92; *p* = 0.789) [19]. Overall, the autoimmune status of bullous disorder did not pose a greater risk of SARS-CoV-2 infection in pemphigus patients [20]. Of note, even the confirmed COVID-19 positive patients with ABID did not develop severe disease-related complications following the exposure to systemic glucocorticoids or immunosuppressants, which is consistent with other autoimmune inflammatory diseases during the pandemic [21,22].

## 3. COVID-19 Vaccine: Trigger or Exacerbate Pemphigus?

In response to the global outbreak, the COVID-19 vaccine emerged as a newly approved product in an effort to confer protective efficacy and prevent the public from infection. Generally, it has been indicated comparatively safe and well-tolerated. In an observational cohort study on vaccine safety which enrolled 21,744 subjects, the reported adverse events primarily include pain at the injection site (>80%), headache (>50%), fatigue (>50%); myalgia, malaise (>40%); pyrexia, chills (>30%), arthralgia (>20%), as well as swelling at the injection site (>10%) [23]. Similarly, another observational cohort study of 2740 subjects reflects that COVID-19 vaccination-related cutaneous adverse responses consisted of angioedema/urticarial rashes (28%), injection-site erythema (24%), generalized pruritus (10%), toxic erythema (8%), erythema multiforme (6%), pityriasis rosea-like eruption (6%) and so on [24]. Notably, a few cases report of new onset or relapse of pemphigus following infusion of COVID-19 vaccines in multiple countries. We reviewed 13 reports of 20 patients, as summarized in Table 1. Actually, the figure is considered an underestimate since data collection is incomplete. Dermatologists and other clinicians should be aware that pemphigus may develop within 3 days to 3 weeks following COVID-19 vaccination, either after the first or second dose injection. On the other hand, dermatologists should fully explain the potential adverse reactions and closely follow up with patients with or without autoimmune bullous skin diseases. Albeit vaccines may trigger or aggravate the pemphigus course, the outcome could be improved with standard routine treatment. In addition, there are certain cases of pemphigus provocation following vaccination in clinic, which could possibly occur after administering vaccine against rabies, influenza, hepatitis B, diphtheria, typhoid, tetanus, and anthrax, etc. [25] The association between onset or reactivation of pemphigus disease and viral infections may be causal, or as a result of drug-induced immunosuppression, or established on the pathogenic connection between viral infection and immune dysregulation leading to autoimmunity. The direct immunopathogenesis link between COVID-19 vaccines and pemphigus has not yet been clarified, but COVID-19 vaccines have proven to be effective against the deadly COVID pandemic. Polack et al. [26] have demonstrated 95% protective efficacy of the two-dose regimen of BNT162b2 conferred in patients aged > 16 years, with remarkable safety and mostly mild-to-moderate adverse events. Similarly, the phase 3 randomized, observer-blinded, placebo-controlled trial of the mRNA-1273 vaccine in 30,420 volunteers showed 94.1% efficacy in preventing COVID-19 illness [27]. These clinical trials provide evidence of the efficacy of COVID-19 vaccines in preventing symptomatic SARS-CoV-2 infection, whereas pemphigus is a rare occurrence in the general population. The vaccines are well tolerated, with mild and easily manageable adverse events in most cases [28]. Based on preliminary studies, pemphigus is not a contraindication to vaccination; and the overall benefit of avoiding severe COVID-19 through vaccination will outweigh the potential risks of disease flare [29].

## 4. The Underlying Biological Mechanism of Vaccine-Induced Pemphigus

Currently, the SARS-CoV-2 vaccines are widely administered to prevent deaths due to COVID-19 infection and to end the pandemic. However, it is reported that only 68% of patients with autoimmune bullous diseases would accept the COVID-19 vaccination without hesitancy or denial, while others showed their concerns of whether the vaccine would induce disease exacerbation or relapse [42]. In contrast, EULAR recommended that live-attenuated vaccines may be considered with caution in patients with rheumatic diseases [43]. Is that the case with the new COVID-19 vaccines in the context of pemphigus? Herein, we attempt to further investigate the direct association between vaccination and pemphigus, in accordance with the classic and emerging pathogenesis of pemphigus, and analyze the efficacy and safety of COVID-19 vaccination in patients with pemphigus. Prior research on vaccine and autoimmunity suggests that molecular mimicry, bystander activation, and epitope spreading along with cytokine cascades, anti-idiotypic network, HLA expression, modification of surface antigens, induction of novel antigens, polyclonal activation of B cells, etc. are involved in both anti-infectious and autoreactivity process [44,45]: (1) Molecular mimicry, which refers to the significant similarity between certain vaccine components and specific proteins in vaccinated subject, has been applied to account for post-vaccination autoimmune phenomena in influenza, hepatitis B, and human papilloma virus vaccines. The similarity between the human protein and vaccine element may lead to immune cross-reactivity [46]. (2) Bystander activation refers to self-epitopes from host damaged tissue stimulating the proliferation of antigen-presenting cells (APCs) along with subsequently autoreactive T cells, with broader activation by cytokines secretion. (3) Following bystander activation, epitope spreading occurs, which refers to the immune responses directed against a different portion of the same protein or a different protein, effectively reducing the possibility that the pathogen escapes the immune mechanism with a single mutation in an immunogenic epitope. The aforementioned immune responses towards the natural infection or artificial antigens (vaccine) may jeopardize the immune system, eventually resulting in an autoimmune disorder. Thus, it is feasible to assume that the COVID-19 vaccine could evoke pemphigus but only when autoreactive T and B cells have to be involved and self-epitopes are exposed. To the best of our knowledge, no research has yet confirmed such mechanisms for vaccine-induced autoimmune disorders. In addition, Martino et al. are convinced that vaccines are not a source of autoimmune diseases [44]. 

The range that COVID-19 vaccines fight against virus mainly includes: nucleic acid (DNA, RNA), protein-based (protein subunit, virus-like particle), viral vector (replicating and non-replicating), and whole virus (attenuated or inactivated) [47]. Sahin et al. performed a cohort clinical trial on BNT162b1, a lipid nanoparticle-formulated nucleoside-modified mRNA that encodes the receptor-binding domain (RBD) of the SARS-CoV-2 spike protein, with the results showing that BNT162b1 promotes robust T cell activity, with RBD-specific CD4+ and CD8+ T cell expansion as well as large amounts of inflammatory cytokines such as IFN-γ, IL-2 and elicit T helper type 1 (TH1)-skewed T cell profile. Concurrently, B cells are also stimulated, with substantial production of plasma cells, memory B cells, and RBD-binding antibodies. The robust T and B immunocytes, accompanied by cytokine cascades, provide the anti-COVID-19 protective efficacy in individuals [11]. Looking back to the pathogeneses of pemphigus, the classic pathogenesis of desmoglein-specific antibodies is apparently incapable to explain the hypothesis about COVID-19 vaccine-induced pemphigus. Another widely-acknowledged pathogenesis of pemphigus is that the disease is partially mediated by classical TH2 cells which produce lineage-specific cytokines such as IL-4 and IL-5. Except for that, the latest research implicates that not only IL-4, but also IL-17 and IL-21 are closely correlated to the onset of autoimmune disorders like pemphigus. That is to say, pemphigus is a TH2 cell-driven disease as well as a complex TH17/TFH17 cell–dominated disease [48]. In the trial of Sahin et al., RBD-specific CD4+ and CD8+ T cells produced IFN-γ, IL-2, or both, but they did not produce IL-4 in the majority of 52 participants, with only one single individual developing a significant IL-4 secretion following BNT162b1 vaccine, indicating the unlikely pathogenic role of a potentially deleterious TH2 responses accounting for vaccine-induced pemphigus [11]. IL-4 production is observed in trials as mentioned following COVID-19 vaccination, whereas data about IL-17 and IL-21 which are viewed as the protective factor of vaccine-induced immunity, are still missing in patients developing pemphigus following COVID-19 vaccine administration [49,50]. Certainly, the etiology of pemphigus and underlying biological mechanisms behind pemphigus and vaccine is still under investigation (Figure 1).

In conclusion, molecular mimicry is caused by genetic similarities of SARS-CoV-2 spike protein components to endogenous cross-reactive human antigens, thereby generating autoreactive lymphocytes and cross-reactive antibodies. The hyper-stimulated state of the immune system plausibly triggers the onset of autoimmune dysregulation in genetically predisposed individuals. The RBD of SARS-CoV-2 is highly immunogenic to mediate detectable antibodies; while the protective capacity of such humoral immune responses remains unclear. On the other hand, vaccine-induced immunity likely has both a cell-mediated and humoral response. Different variables including neutralization antibodies and memory cells, as well as the potential emergence of variants, may also impact the responses of patients with immune-mediated inflammatory diseases (IMID) to vaccination [29,51]. However, it has not yet been clarified whether vaccines should be held accountable for pemphigus. Given the risks of hospitalization and death associated with COVID-19, vaccination still ought to be advocated in most patients.

## 5. Rituximab or Not for Pemphigus Patients?

In recent decades, biologics and new target synthetic agents have tremendously improved the clinical outcomes of autoimmune diseases. However, the COVID-19 pandemic raised concerns that immunosuppressive agents may pose a risk of increased severity and duration of COVID disease since they could lead to the depletion of B-cell lymphocyte counts. Rituximab, is a monoclonal anti-CD20 antibody as first-line therapeutic strategy for pemphigus [52]. Prolonged B-cell depletion, B-cell–T-cell crosstalk, and neutropenia are considered the possible mechanisms of increased susceptibility of COVID-19 infection following Rituximab infusion [53]. In a recent study, Joly P and colleagues [54] reported a more than 5-fold higher risk of COVID-19 infection in patients with AIBD who received Rituximab compared with those who did not. Similarly, in the Swedish, Spelman et al. [55] conducted research on 476 COVID-19 cases who had formerly developed multiple sclerosis, and found Rituximab in confirmed COVID-19 patients was associated with 2.95 times the odds of hospitalization relative to other combined therapies. Additionally, in an analysis of 3729 patients with rheumatic diseases [56], most disease-modifying anti-rheumatic drugs (DMARDs) were not associated with higher odds of COVID-19-related death, except for rituximab. The results indicate that Rituximab is a major risk factor for COVID-19 infection. Avouac et al. [57] reported three COVID-19-associated patients with systemic sclerosis who routinely undergone Rituximab experienced late clinical deterioration (up to day 23) to severe pneumonia. The possible explanation is that the presentation of coronavirus antigens might be impaired by Rituximab and the activation of immune cells consequently delayed, holding up the onset of the cytokine storm. Therefore, long-term follow-up is required for AIBD patients treated with Rituximab. Opinions on the safety of RTX in COVID-19 are controversial. Paradoxically, a small case series observing 132 patients with AIBDs, 52 of whom had a history of Rituximab treatment, found there is no significant difference in the rates of SARS-CoV-2 positivity between patients with AIBD immunosuppressed by Rituximab and those not on Rituximab (9.1% vs. 12.1%) [58]. In a retrospective study involving 704 ABID patients, Mahmoudi et al. [59], purposed such risks would gradually decrease after the last dose of Rituximab. Uzuncakmak et al. [60] also found that there is no significant correlation between the lymphocyte count and the number of Rituximab treatment courses received, indicating that the risk of COVID-19 infection is independent of the received Rituximab infusion. A multicenter study from Turkey [61] (PMID: 35243732) which enrolled 247 patients reported that none of the patients treated with Rituximab required intensive care or died of COVID-19 infection, and made a conclusion that Rituximab did not require additional safety cautions relative to other immunosuppressives. Theoretically, the crucial immunocyte for which Rituximab yields its efficacy is CD20-expressing B cells whereas the coronavirus primarily attacks T cells. Chen et al. [62] analyzed the immunological characteristics of 21 subjects who are classified as severe and moderate COVID-19 cases, and found the absolute numbers of CD4+ and CD8+T lymphocytes profoundly reduced in almost all patients; particularly much lower in severe cases compared with moderate cases. Conversely, the proportion of B lymphocytes was not detected with a decrease. As a result, the use of Rituximab ought not to pose a considerable threat for pemphigus patients to develop COVID-19 infection. As such, Baker et al. [63] proposed that drug-induced B-cell subset inhibition would not have an adequate impact on innate and CD8 T-cell responses, which are crucial immune responses against SARS-CoV-2. From our perspective, individual heterogeneity and comorbidities are potential factors for virus outbreaks. Considering the interplay of the complicated pathophysiological mechanisms including clinical features of autoimmune disease, comorbidities, viral aggression, and immune responses to coronavirus, each patient might be a specific case when confronted with COVID-19 [64]. To date, data concerning Rituximab during the pandemic are still insufficient, and whether Rituximab is associated with a specific risk of more severe COVID-19 is not yet established. Physicians should consider the risk-benefit ratio of individual cases, and Rituximab ought to be administered with caution during the COVID-19 epidemic. 

## 6. Immune Responses to COVID-19 Vaccines in Rituximab Treatment

With vaccines and Rituximab becoming widely available for pemphigus patients, the immune responses to SARS-CoV-2 vaccines in patients are undefined. Studies have found that pemphigus patients following Rituximab develop feeble humoral responses to vaccination, which may not correlate directly with protection from infection [29]. In April 2021, Deepak et al. [65] assessed the two-dose vaccine immunogenicity in 133 patients developing chronic inflammatory conditions and 53 immunocompetent controls, and found that glucocorticoids and B cell depletion therapy could more severely impede optimal responses. In other studies conducted by Boyarsky et al. [66] and Spiera et al. [67], patients treated with Rituximab were less likely to develop an antibody response to the SARS-CoV-2 mRNA vaccine, as was observed for influenza [68] and pneumococcal vaccination [69]. Patients receiving regular Rituximab have failed to mount any detectable antibody response, even with a 6-month interval between Rituximab dosing and vaccination [70]. Based on the available data, the chief concern for patients with immune-mediated inflammatory diseases, including pemphigus, appears to be the efficacy of the immune responses these patients will be able to generate after receiving Rituximab. Patients treated with Rituximab may require an extra vaccine booster to enable reasonable antibody responses following mRNA-based SARS-CoV-2 vaccination [37]. However, as no cutoff titer that is most strongly associated with sufficient protection has been defined, the effect of reduced antibody levels on protection remains unclear. Additionally, patients receiving biologics may produce diminished immune responses, it is not yet clear what the clinical significance of this altered immune response may actually be. These data do not directly evaluate protection from SARS-CoV-2 infection nor prevention of hospitalization. Therefore, it is crucial to follow up with these patients after vaccination to determine the safety, efficacy, and duration of the vaccine.

## 7. The Management Strategies of Pemphigus in COVID-19 Pandemic 

Dermatology is a visual specialty that is particularly suitable for telemedicine, which is conducted on a virtual platform to remotely deliver health information such as diagnoses or consultations and perform internet-based management as well as social care [71]. Given that social distancing is one of the most effective safeguards to limit the spread of the virus, it is recommended that mild or stable cases of pemphigus patients could adopt teledermatology for following-up so as to minimize patient referrals to healthcare centers [72]. Glucocorticoids and steroid-sparing immunosuppressants ought to be tapered to the minimum possible necessary dose. For mild pemphigus (PDAI ≤ 15), topical corticosteroids could be applied alone or combined with a systemic corticosteroid treatment at the dose of prednisone 0.5–1.0 mg/kg/day [8]. Specifically, 10 mg daily of prednisone-equivalent is the common permitted steroid dose within clinical studies [73], on account of ≥20 mg of prednisone being regarded as a usual cutoff threshold to present its immunosuppressive properties [14]. In the circumstances that patients are in a stable condition, the cessation of the immunosuppressants is not recommended, since interruption may mediate dysregulation of inflammatory cytokines, which may possibly exacerbate pemphigus activity, resulting in an increased risk of viral infection and mortality [74]. The disease inflammation control by means of immunosuppressants is likely to reduce aggressive organic responses to SARS-CoV-2. Hospitals should vigorously promote online medical care and provide drug mailing services for patients with stable conditions. Doctors must take advantage of the online platform to strengthen communication with patients, closely monitor the changes in patients’ conditions, and carefully adjust the treatment regimen [75].

All patients with moderate-to-severe pemphigus admitted to the hospital are performed COVID-19 swabs, and immunosuppressive medications would be appropriately administered for disease control. Rituximab has been approved as a first-line treatment for moderate-to-severe pemphigus in updated guidelines [52]. Nevertheless, the optimal rituximab timing and dosing strategies for induction and maintenance of remission remain unclear. Other adjuvant immunosuppressants such as mycophenolate mofetil (MMF), azathioprine (AZA), and cyclophosphamide (CTX), are recommended as second therapeutic choices in clinic [76].

Patients who receive immunosuppressants such as Rituximab represent a potentially susceptible population requiring the constant monitoring of adverse effects, particularly bacterial, fungal or viral infection. Given that the median incubation period of COVID-19 is around 4 to 5 days and can take up to 1 month, with a peak around day7 to day10 [77], it is advisable to postpone the Rituximab dose for at least 30 days of infection onset (based on potential length of viral shedding) for those pemphigus patients with confirmed or suspected COVID-19 infection [76]. Intravenous immunoglobulin is the safest alternative to systemic corticosteroids for pemphigus of all ages, and it has been proposed as a potential option for COVID-19 [78,79]. 

Emotional management is as important as disease management for patients with pemphigus. The first Chinese dermatologists to face SARS-CoV-2 have shared their experiences about performing telemedicine on 38 AIBD patients during the worst three months of the epidemic in China [80]. It was found that the majority of patients were confronted with discontinuation of treatment and anxiety, among whom 17 stopped their medication in the absence of consultation with dermatologists, mainly owing that prescription drugs are not available for purchase during the epidemic or that patients themselves reduced or stopped drugs for concerning side effects. Twenty-one patients suffer from anxiety since media reports say patients with chronic diseases are susceptible to COVID-19 infection which brought patients a great psychological burden. Therefore, it is necessary for doctors to provide medication guidance and psychological counseling for patients. On the other hand, pandemic-related emotional stress, as well as nonstandard withdrawal of immunosuppressants for fear of COVID-19 predisposition risk might be considered exacerbating factors for pemphigus [81]. Hence, strict adherence to health principles and expert protocols while avoiding emotional stress may help prevent exacerbation or recurrence of pemphigus (Figure 2). Albeit the expert guidance could provide a credible reference for clinical practice, the specific treatment strategy should be established relying on risk-benefit profiles for each individual. For this particularly vulnerable patient group, surveillance and special precautions must remain, and a tapering course of corticosteroids to minimize susceptibility to life-threatening infections is required.

## 8. Discussion

Management of pemphigus in the COVID-19 epidemic is a challenge for dermatologists. Overall, basic infection-prevention principles should be strictly adhered and drug-induced immunosuppression should be minimized as possible in treating pemphigus patients [82]. As the COVID-19 vaccine has become widely distributed, surveillance of safety issues related to these vaccines is progressing, vaccination should not be hindered through false beliefs about the extension or flare-ups, the prevalence of adverse events, as well as inefficient immunization, so that critical or potentially fatal adverse reactions could be safely avoided [83]. Due to the limited data about the diagnosed pemphigus following the COVID-19 vaccine in clinic, it is incapable to draw a definite conclusion that the COVID-19 vaccine may trigger the onset, exacerbation, or relapse of pemphigus thus far. With regard to the summarized cases of pemphigus onset or aggravation after receiving the COVID-19 vaccine, we can not exclude that some cases which are probably associated with personal characteristics are coincidental. Notably, uncontrolled infection-induced autoreactive responses may convert into autoimmune disorders in genetically strongly susceptible individuals [84]. In conclusion, whether COVID-19 vaccines are truly or commonly related to an increased risk of pemphigus has yet to be delineated. Albeit the COVID-19 vaccine may induce potentially autoimmune manifestations in certain genetically predisposed individuals, it is undeniable that it can also strengthen our immunity and prevent us from virus infection or restrict virus replication. According to the interim guidelines from the US CDC Advisory Committee on Immunization Practices (ACIP), it is recommended that immunocompromised patients without contraindications to vaccination should be immunized and authorized vaccines against COVID-19, despite that the immune responses to COVID-19 vaccines may be attenuated. Generally, it is preferred to vaccinate patients while in remission or prior to planned immunosuppression. In the case of Rituximab, it is better to initiate the entire vaccination series ≥ 4 weeks prior to the Rituximab cycle or 12–20 weeks after completion of a Rituximab cycle, but the optimal time points are not clearly defined [85]. Better communication of information with patients is needed to achieve peak immunization, while keeping in mind the necessary precautions and clarifying contraindications so that high-risk patients can safely avoid any serious, severe adverse reactions, particularly those elderly patients with coexisting morbidities.

It was demonstrated that SARS-CoV-2 vaccines do not pose a more prominent danger than virus infection themselves [86]. For those pemphigus patients who have received Rituximab or not, the COVID-19 vaccination could be administered. In specific circumstances, expert judgement on vaccination options including the type, infusion interval, as well as proper Rituximab doses are required. In the coming future, an algorithm or scoring system, to predict the likelihood of vaccine-associated reaction is expected to establish. Importantly, the question of whether vaccines can induce autoimmune diseases or increase the prevalence in susceptible individuals and how to strike a balance between disease control with a minimum of immunosuppression to reduce the risks of COVID-19 adverse effects might be the focus of the future research. Novel vaccine development paradigms, as well as personalized therapeutic strategies, should be necessitated. Dermatologists should evaluate the benefit-to-risk ratio on a case-by-case basis. Caution could be focused on avoiding exposure to high-risk contacts of infected people. As well, it is essential to advise the public to pay more attention to hand hygiene and social distancing. For the time being, it is unnecessary to modify the present vaccination recommendations for patients with pemphigus. The overall benefits of vaccines outweigh the likelihood of instigating autoimmune manifestations, which still require further empirical data with COVID-19 vaccines in the context of pemphigus patients in the real world. To gain knowledge on COVID-19 vaccines and pemphigus, clinical trials can be initiated to compare immune responses to COVID-19 vaccines in autoimmune diseases requiring immunosuppressive medications, including pemphigus patients. Also, the high-quality, large-scale, population-based, longer-term observation studies are much more persuasive and convincing than anecdotal cases, case reports, and uncontrolled observational studies. Efforts on how COVID-19 vaccines trigger the immune responses in the human body, and the novel pathogenesis of pemphigus should be on the way to draw a definite conclusion on this question with an aim to ensure vaccine safety and efficacy and provide better management guidance for patients in pemphigus during the COVID-19 epidemic.

## Figures and Tables

**Figure 1 jcm-11-03968-f001:**
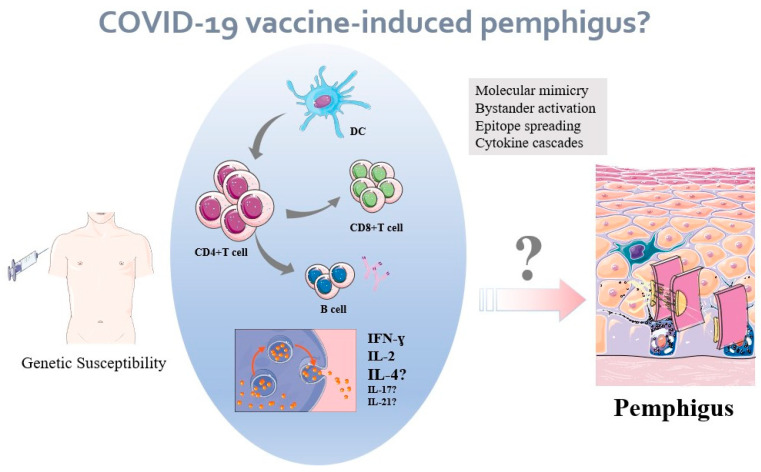
An unproven hypothesis: mechanisms of COVID-19 vaccine-induced pemphigus? It is hypothesized that vaccination against COVID-19 could trigger an immunological response in genetically predisposed individuals. The vaccine-related autoimmune responses may include molecular mimicry, bystander activation, and epitope spreading, etc. The similar components of human protein promote dendritic cell maturation and elicit robust T and B cell responses, thereby subsequently activating bystander autoreactive lymphocytes and episode spreading, along with large amounts of cytokines, concurrently contributing to the autoimmune inflammation in pemphigus patients.

**Figure 2 jcm-11-03968-f002:**
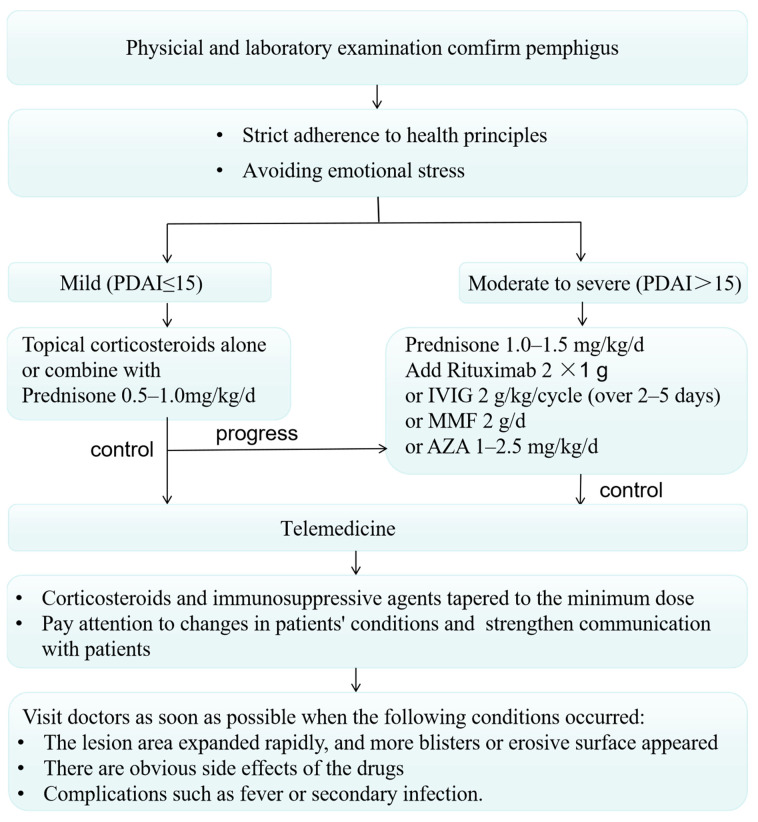
Flowchart of management strategies for pemphigus during the epidemic. PDAI, Pemphigus Disease Area Index; IVIG, intravenous immunoglobulin; MMF, mycophenolate mofetil; AZA, azathioprine.

**Table 1 jcm-11-03968-t001:** Summary of pemphigus onset/flare following SARS-CoV-2 vaccination in published studies.

Study	Age	Sex	Vaccine	Technology/Platform	Dose of Vaccine	Diagnosis	New Onset or Flare	Latency
Solimani et al., 2021 [30]	40	F	Pfizer/BioNTech	mRNA	1	PV	New onset	5 days
Damiani et al., 2021 [31]	40	M	Moderna	mRNA	1	PV	flare	3 days
	80	M	Pfizer/BioNTech	mRNA	1	PV	flare	3 days
Thongprasom et al., 2021 [32]	38	F	AstraZeneca	Modified chimpanzee adenovirus (ChAdOx1)	1	Oral pemphigus	New onset	1 week
Koutlas et al., 2021 [33]	60	M	Moderna	mRNA	2	PV	New onset	7 days
Saleh et al., 2022 [34]	35	F	Sinopharm	Inactivated SARS-CoV-2 (Vero cells)	2	PV	flare	5 days
Singh et al., 2022 [28]	44	M	AstraZeneca	Modified chimpanzee adenovirus (ChAdOx1)	2	PV	New onset	1 week
Akoglu et al., 2022 [35]	69	F	Sinovac	Inactivated SARS-CoV-2	2	PV	New onset	1 week
	58	F	Sinovac	Inactivated SARS-CoV-2	2	PV	flare	-
	31	F	Pfizer/BioNTech	mRNA vaccine	1	PV	flare	1 week
Hali et al., 2022 [36]	50	F	Pfizer/BioNTech	mRNA	1	PF	New onset	15 days
	58	F	Pfizer/BioNTech	mRNA	1	PV	New onset	1 month
Lua et al., 2022 [37]	83	M	Pfizer-BioNTech	mRNA	2	PF	New onset	2 days
Calabria et al., 2022 [38]	60	F	Pfizer-BioNTech	mRNA	2	PV	New onset	7 days
Knechtl et al., 2022 [39]	89	M	Pfizer/BioNTech	mRNA	2	PV	New onset	1 month
Yıldırıcı et al., 2022 [40]	65	M	Pfizer/BioNTech	mRNA	1	PF	New onset	3 weeks
Hatami et al., 2022 [41]	34	M	Astrazeneca	Modified chimpanzee adenovirus (ChAdOx1)	1	PV	New onset	a few days
	61	M	Astrazeneca	Modified chimpanzee adenovirus (ChAdOx1)	1	PV	Flare	about 1 week

M, male; F, female; PV, pemphigus vulgaris; PF, pemphigus foliaceus.

## Data Availability

Not applicable.

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
