# Peer review of "Pemphigus during the COVID-19 Epidemic: Infection Risk, Vaccine Responses and Management Strategies"

_jcm, 2022, doi:10.3390/jcm11143968_

Round 1

Reviewer 1 Report

The paper should be more organized and the mechanistic association between COVID-19 and pemphigus need to be added. There are lack of clearly conclusions. Moreover, it should be discussed if benefits of vaccination against COVID-19 outweigh risks of developing pemphigus.  

Author Response

Thank you for your letter and  comments concerning our manuscript entitled “Pemphigus during the COVID-19 epidemic: infection risk, vaccine responses and management strategies” (jcm-1756138). We would like to thank the reviewers for thoroughly reviewing our manuscript and providing insightful comments that helped us to significantly improve the quality of the manuscript. We have studied comments carefully and have made correction which we hope meet with approval. The reviewer comments are laid out below in italicized font. Revised portion are marked in red in the PDF. Thanks again.

Reviewer #1: The paper should be more organized and the mechanistic association between COVID-19 and pemphigus need to be added. There are lack of clearly conclusions. Moreover, it should be discussed if benefits of vaccination against COVID-19 outweigh risks of developing pemphigus.  

Response): Thank you for your suggestions. We have re-structured the discussion section of our manuscript as much as possible according to the comments; and we also rectify other parts in the passage to make this paper more organized. We endeavored to search for all the relevant articles, whereas COVID-19 is a relatively new topic in recent years and there are few studies describing their potential mechanisms between infection and pemphigus status. Most literature merely reported the objective outcomes of trials or cases, while none have a deeper exploration from mechanic aspect. Moreover, the mechanisms of vaccine-induced-pemphigus involving molecular mimicry, bystander activation, etc. have been basically presented in the original manuscript.

Hopefully this passage can also help shed light on the complicated interplay might in further studies and their immunological underlying mechanisms could be elucidated more explicitly. A summary of the underlying biological mechanism of vaccine-induced pemphigus was added to the article in line 204-216. As suggested, we have now added a paragraph in discussing the benefits of getting vaccinated against COVID-19 outweigh the risks of developing pemphigus in line 119-138. Please let us know if you have questions or additional suggestions.

Reviewer 2 Report

This is an important topic. In general, I agree with the authors that RTX is still appropriate to use in AIBD particularly pemphigus given its efficacy and more targeted immunosuppression. However RTX has certainly been shown to be associated with increased risk of COVID in some studies of AIBD pts and increased risk of mortality in rheumatologic patients (see below). Therefore, I would argue that it is not appropriate to state "concerns are unnecessary" and the authors seem to understate potential risks while overstating the certainty of no risk.

Would recommend the authors include the following studies demonstrating increased risk of COVID in setting of RTX use in AIBD.
- J Am Acad Dermatol. 2021 Apr;84(4):1098-1100.
- J Am Acad Dermatol. 2022 Feb;86(2):494-497.

Additionally, would recommend the authors review and consider including this article looking at rates of COVID related death in a rheumatologic population:
- Ann Rheum Dis. 2021 Jul;80(7):930-942.

Additional comments:

- line 229 - I would not say that the prevalent viewpoint is to postpone RTX during the pandemic; in fact, for those who suggested that in the very early days of the pandemic, it was argued against (J Am Acad Dermatol. 2021 Jan;84(1):e59-360) and as the pandemic dragged on, quickly become clear that delay was not prudent. In the JAAD expert guidelines (J Am Acad Dermatol. 2020 Oct;83(4):1150-1159.) it was recommended to avoid RTX with exception of AIBD.
- lines 235 - 237 -  "these drugs would confer a protective role of interrupting viral replication in the advanced severe phase of COVID infection." --> this is theoretical, should not be stated as fact but as a theory and citation should be included
- lines 281-285 - I don't think that one study along can "indicate that immunosuppressive agents and Rituximab can be used with precaution during the pandemic". In my opinion, this entire article really should be framed as a delicate balance between potential risks to patients of which there are conflicting studies but overall thought to confer benefit given efficacy of RTX and we should try to reduce risks of COVID as much as able with vaccines, pre-exposure ppx such as Evusheld, early treatment such as Paxlovid.
- page 9 figure - confirm is spelled incorrectly, prednisone is spelled incorrectly; for mild PDAI is systemic prednisone necessary for all? consider topical steroids? conversely even mild PDAIs can have significant QOL effects and may require RTX  or other therapies. I don't think it's necessarily black and white between mild and mod/severe PDAI. I think for all patients it should be a discussion of risk/benefit. I try to avoid prednisone if I don't need it, prednisone also can increase risk of COVID.
- Would recommend to add a section on efficacy of vaccines in rituximab, we know that the vaccine effect is reduced after receiving RTX, please comment on this

Grammar comments:

- line 82 - should be "were diagnosed" not "got diagnosed"
- line 103 - I'm not sure what you mean when you say "COVID-19 vaccine developed"
- line 117 should be "either after the first" not "either the first"
- line 122 I don't think you need to restate pemphigus incidence as you said earlier
- lines 124-127 - unclear when you say "not rarely seen", are you saying this is common? please rewrite
- line 141 "prior researches" should be prior research 
- lines 145 to 158 - what are the 1, 2, 3 you are listed? are these 3 different mechanisms, 3 subsequent mechanisms, etc please clarify with a sentence explaining prior to listing them
- lines 163 - what is an off the shelf vaccine? this sentence strangely worded, please clarify this sentence
-line 175 - specific is spelled incorrectly
-line 175 - what classic pathogenesis? can rbd abs perform epitope spread to dsg?
- line 176 - you start to discuss here the pathogenesis of pemphigus in the same paragraph you are discussing how vaccines target viruses; this is confusing, consider re-ordering for improved flow and clarity or starting a new paragraph with an opening statement to explain your transition
- Your figure is very helpful, would recommend improving preceding paragraphs as above
- line 224 - should be "leads to"
- line 348 - should be 'dermatologists"
- line 349 - "on not exposed to high-risk" doesn't make sense, better to say something like "caution could be focused on avoiding exposure to high risk contacts or infected people"
- lines 361 - "novel pathogenesis of pemphigus" or of vaccine-induced pemphigus?

Author Response

Thank you for your letter and comments concerning our manuscript entitled “Pemphigus during the COVID-19 epidemic: infection risk, vaccine responses and management strategies” (jcm-1756138). We would like to thank the reviewers for thoroughly reviewing our manuscript and providing insightful comments that helped us to significantly improve the quality of the manuscript. We have studied comments carefully and have made correction which we hope meet with approval. The reviewer comments are laid out below in italicized font. Revised portion are marked in red in the PDF. Thanks again.

Reviewer #2: Overall comments:

This is an important topic. In general, I agree with the authors that RTX is still appropriate to use in AIBD particularly pemphigus given its efficacy and more targeted immunosuppression. However, RTX has certainly been shown to be associated with increased risk of COVID in some studies of AIBD pts and increased risk of mortality in rheumatologic patients (see below). Therefore, I would argue that it is not appropriate to state "concerns are unnecessary" and the authors seem to understate potential risks while overstating the certainty of no risk.
Response 1) Sincerely thanks very much for this helpful review - we appreciate the time you have taken to carefully read and provide constructive feedback on the article.

We completely agree that our original wording was inappropriate and have substantially revised the manuscript, including the abstract, in accordance with the comments. Line 11-17.

Would recommend the authors include the following studies demonstrating increased risk of COVID in setting of RTX use in AIBD.
- J Am Acad Dermatol. 2021 Apr;84(4):1098-1100.
- J Am Acad Dermatol. 2022 Feb;86(2):494-497.

Response 2) Thanks for your suggestion. We feel much appreciated that you recommended an up-to-date reference for us. We have added this reference in the review. Mainly on the Line: 252-253 and 233-235

Additionally, would recommend the authors review and consider including this article looking at rates of COVID related death in a rheumatologic population:
- Ann Rheum Dis. 2021 Jul;80(7):930-942.

Response 3) Thanks for your comments. We have added this reference in line 238-241.

line 229 - I would not say that the prevalent viewpoint is to postpone RTX during the pandemic; in fact, for those who suggested that in the very early days of the pandemic, it was argued against (J Am Acad Dermatol. 2021 Jan;84(1):e59-360) and as the pandemic dragged on, quickly become clear that delay was not prudent. In the JAAD expert guidelines (J Am Acad Dermatol. 2020 Oct;83(4):1150-1159.) it was recommended to avoid RTX with exception of AIBD.

Response 4) Thank you very much for your comments and this is a very controversial issue. During the COVID-19 epidemic wave, there are two main opposing views on whether RTX would increase the risk of COVID-19 infection. We have made detailed discussion in the manuscript (Line 233-267). In the early days of the pandemic, the EADV (European Academy of Dermatology and Venereology) AIBD task force currently did not recommend the use of RTX to prevent relapse, especially for non-vaccinated patients (PMID: 33655539). According to our clinical experience, RTX is still the first-line therapy for severe pemphigus, without an increase risk of COVID-19 infection. Data is being compiled and not published. Therefore, we fully agree with the editor that it is unwise to delay using RTX. As for the timing use of RTX, we mentioned in the article (Line 392-394), but did not discuss it in detail as the focus. However, when pemphigus patients with confirmed COVID-19 infection, it is advisable to postpone the Rituximab treatment.

Kindly note that the article you mentioned (J Am Acad Dermatol. 2021 Jan;84(1):e59-360) was not found. If there is any question in the revised version of our manuscript, please feel free to contact us and we will make our best efforts to meet your requirements.

lines 235 - 237 -  "these drugs would confer a protective role of interrupting viral replication in the advanced severe phase of COVID infection." --> this is theoretical, should not be stated as fact but as a theory and citation should be included

Response 5) Thank you for your professional advice. This viewpoint is extracted on our own after browsing plenty of literature. Pitifully, we couldn’t find the literature which definitely propose this viewpoint. Perhaps it is kind of subjective. Thank you for your reminding. Here, we have removed this improper sentence in the passage.

lines 281-285 - I don't think that one study along can "indicate that immunosuppressive agents and Rituximab can be used with precaution during the pandemic". In my opinion, this entire article really should be framed as a delicate balance between potential risks to patients of which there are conflicting studies but overall thought to confer benefit given efficacy of RTX and we should try to reduce risks of COVID as much as able with vaccines, pre-exposure ppx such as Evusheld, early treatment such as Paxlovid.

Response 6) Thank you so much for your comments and valuable advices. We have made detailed discussions and revised this paragraph in the manuscript, and have removed "indicate that immunosuppressive agents and Rituximab can be used with precaution during the pandemic"  (Line 248-251).

 page 9 figure - confirm is spelled incorrectly, prednisone is spelled incorrectly; for mild PDAI is systemic prednisone necessary for all? consider topical steroids? conversely even mild PDAIs can have significant QOL effects and may require RTX  or other therapies. I don't think it's necessarily black and white between mild and mod/severe PDAI. I think for all patients it should be a discussion of risk/benefit. I try to avoid prednisone if I don't need it, prednisone also can increase risk of COVID.

Response 7) Firstly, we are sorry for our mistakes and thank you for your reminder. Secondly, we thank the editor for this guidance and have now modified the Figure2, as directed. We also added some statements on line 313-315 and 359-361. There are 2 clinical scores, the Pemphigus Disease and Area Index (PDAI) and/or Autoimmune Bullous Skin Intensity and Severity Score (ABSIS), which are currently being used as clinical outcome parameters and in clinical trials for the evaluation of the extent and activity of pemphigus.

Last but not least, we would like to thank the reviewer for his/her interesting questions of “prednisone also can increase risk of COVID. Data evaluating the use of systemic corticosteroids in COVID-19 are controversial. Existing safety data indicate that daily doses of systemic corticosteroids are associated with an increased susceptibility to overall infection (PMID: 33504483). However, preliminary evidence from a large-scale RCT suggests that the use of dexamethasone reduces mortality in hospitalized COVID-19 patients (DOI:10.1056/NEJMoa2021436). A subsequent meta-analysis about performing glucocorticoids on critically ill patients with COVID-19, reported that 28-day all-cause mortality was lower among patients who received corticosteroids compared with those who received usual care or placebo (PMID: 32876694). It is likely that the beneficial effect of glucocorticoids in severe viral respiratory infections is dependent on the selection of the right dose, at the right time, in the right patient. Indeed, prednisone also increases the risk of COVID-19, but prednisone can help severely ill patients recover at the same time. Further data are required to define the optimal use and potential complications of systemic corticosteroid in the setting of COVID-19.

Would recommend to add a section on efficacy of vaccines in rituximab, we know that the vaccine effect is reduced after receiving RTX, please comment on this

Response 8) Thanks for your advisable comments to make this review more completed and fulfilling. Here we add a section “Immune responses to COVID-19 vaccines in Rituximab treatment” in the new version to illustrate diminished immune responses to vaccination in RTX. (Line 280-304)

Grammar comments:

Response 9) Thank you for pointing out these problems. We are very sorry for making the mistakes. We have carefully proofread the whole manuscript and make the revisions as suggested.

- line 82 - should be "were diagnosed" not "got diagnosed" 

 Thank you for your comments. Revised as suggested. (Line 81)

- line 103 - I'm not sure what you mean when you say "COVID-19 vaccine developed"

Thank you for your comments. We have altered the word “developed” to “emerged” to make the sentence more understandable. (Line 102)

- line 117 should be "either after the first" not "either the first"

Thank you for your comments. Revised as suggested. (Line 116-117)

- line 122 I don't think you need to restate pemphigus incidence as you said earlier

Thank you for your comments. We have removed this sentence in the new version of manuscript.

- lines 124-127 - unclear when you say "not rarely seen", are you saying this is common? please rewrite

 Thank you for your comments. We have rewritten this sentence in the new version of manuscript. (Line 119-123)

- line 141 "prior researches" should be prior research 

 Thank you. We have made the revisions as suggested. In addition, we also revised some other changes in the review: mainly on line 40, 88, 155 - “researches” to “research”

- lines 145 to 158 - what are the 1, 2, 3 you are listed? are these 3 different mechanisms, 3 subsequent mechanisms, etc please clarify with a sentence explaining prior to listing them

Thank you for your question. What we would like to express is that such 3 hypothesized mechanisms are involved in vaccine-induced immunity. They also act as the 3 chief mechanisms which emerge and have some illustrations in most literature. They are 3 different mechanisms; the hypothesis indeed reveals they may occur in order. The intention we list 1, 2, 3 is to make the expression more clearly; the reference articles also have similar descriptions which listed 1, 2, 3. And we also illustrated in the former sentence, “Prior research on vaccine and autoimmunity suggest that molecular mimicry, bystander activation, and epitope spreading along with cytokine cascades, anti-idiotypic network, HLA expression, modification of surface antigens, induction of novel antigens, polyclonal activation of B cells, etc. are involved in both anti-infectious and autoreactivity process”. We feel appreciated for your suggestion, but we would like to express our sincere sorry we thought there is no need to add another sentence. Would you still feel puzzled after reading our reply? If so, please let us know and we would make further corresponding revision. Thank you.

- lines 163 - what is an off the shelf vaccine? this sentence strangely worded, please clarify this sentence

Thank you for your advice. Off-the-shelf means ready-made, or the present vaccines on the market. In order not to cause misunderstanding, we have removed this word in the passage.

-line 175 - specific is spelled incorrectly

Thank you for your comments. Revised as suggested. (Line 187)

-line 175 - what classic pathogenesis? can rbd abs perform epitope spread to dsg?

Thank you for your question. The classic pathogenesis we want to expressed is that IgG autoantibodies, which directed against the desmosomal adhesion proteins, mainly desmoglein 3 and desmoglein 1 in epidermal keratinocytes, cause the blisters and epidermal splitting due to the compromised cell adhesion. And there is no literature reported that vaccines could mediate such immune responses.

- line 176 - you start to discuss here the pathogenesis of pemphigus in the same paragraph you are discussing how vaccines target viruses; this is confusing, consider re-ordering for improved flow and clarity or starting a new paragraph with an opening statement to explain your transition
- Your figure is very helpful, would recommend improving preceding paragraphs as above

Thank you for your suggestion. Originally, we intent to explain the hypothesis from the aspect of existing pathogenesis of pemphigus. This is why we illustrate such sentences about pathogenesis of pemphigus. The classic pathogenesis about desmoglein-specific antibodies is obviously incapable to explain the hypothesis about COVID-19 vaccine-induced pemphigus; therefore, we come to another pathogenesis about the cytokines.

- line 224 - should be "leads to"

Thank you for your comments. We have revised this paragraph in the paper and removed this grammar mistake.

- line 348 - should be 'dermatologists"

Thank you for your comments. Revised as suggested. (Line 409)

- line 349 - "on not exposed to high-risk" doesn't make sense, better to say something like "caution could be focused on avoiding exposure to high risk contacts or infected people"

Thank you for your comments. Revised as suggested. (Line 410-411)

- lines 361 - "novel pathogenesis of pemphigus" or of vaccine-induced pemphigus?

Thank you for your careful question. The expression of “the novel pathogenesis of pemphigus should be on the way” is what we want to express. From our perspective, the expression of "novel pathogenesis of vaccine-induced pemphigus” is kind of improper, since vaccine-induced pemphigus is only a hypothesis, not a proven fact. If the research about the novel pathogenesis of pemphigus develops, perhaps it would provide some insights for this hypothesis.

Lastly, we would like to express our sincere gratitude for what you have done for us. We sincerely hope our manuscript will meet with approval. Please let us know if you have any other questions or additional suggestions. Thanks again.
